# Inconsistency in Shoulder Arthrometers for Measuring Glenohumeral Joint Laxity: A Systematic Review

**DOI:** 10.3390/bioengineering10070799

**Published:** 2023-07-04

**Authors:** Eluana Gomes, Renato Andrade, Cristina Valente, J. Victor Santos, Jóni Nunes, Óscar Carvalho, Vitor M. Correlo, Filipe S. Silva, J. Miguel Oliveira, Rui L. Reis, João Espregueira-Mendes

**Affiliations:** 1Clínica Espregueira—FIFA Medical Centre of Excellence, 4350-415 Porto, Portugal; 2Dom Henrique Research Centre, 4350-415 Porto, Portugal; 3Porto Biomechanics Laboratory (LABIOMEP), Faculty of Sports, University of Porto, 4200-450 Porto, Portugal; 4Centre for Microelectromechanical Systems (CMEMS-UMINHO), Campus Azurém, University of Minho, 4800-058 Guimarães, Portugal; 5Serviço de Ortopedia e Traumatologia do Hospital de Santa Maria Maior, 4750-333 Barcelos, Portugal; 6School of Medicine, University of Minho, 4710-057 Braga, Portugal; 7LABBELS Associate Laboratory, University of Minho, 4800-058 Guimarães, Portugal; 8ICVS/3B’s–PT Government Associate Laboratory, 4805-017 Guimarães, Portugal; 93B’s Research Group, I3Bs—Research Institute on Biomaterials, Biodegradables and Biomimetics Headquarters of the European Institute of Excellence on Tissue Engineering and Regenerative Medicine, AvePark, Parque de Ciência E Tecnologia, University of Minho, Zona Industrial da Gandra, Barco, 4805-017 Guimarães, Portugal; 10Pro2B, Consultoria e Gestão de Projetos, AvePark—Parque de Ciência e Tecnologia, Zona Industrial da Gandra, Barco, 4805-017 Guimarães, Portugal

**Keywords:** shoulder, glenohumeral, arthrometer, laxity, stiffness

## Abstract

There is no consensus on how to measure shoulder joint laxity and results reported in the literature are not well systematized for the available shoulder arthrometer devices. This systematic review aims to summarize the results of currently available shoulder arthrometers for measuring glenohumeral laxity in individuals with healthy or injured shoulders. Searches were conducted on the PubMed, EMBASE, and Web of Science databases to identify studies that measure glenohumeral laxity with arthrometer-assisted assessment. The mean and standard deviations of the laxity measurement from each study were compared based on the type of population and arthrometer used. Data were organized according to the testing characteristics. A total of 23 studies were included and comprised 1162 shoulders. Populations were divided into 401 healthy individuals, 278 athletes with asymptomatic shoulder, and 134 individuals with symptomatic shoulder. Sensors were the most used method for measuring glenohumeral laxity and stiffness. Most arthrometers applied an external force to the humeral head or superior humerus by a manual-assisted mechanism. Glenohumeral laxity and stiffness were mostly assessed in the sagittal plane. There is substantial heterogeneity in glenohumeral laxity values that is mostly related to the arthrometer used and the testing conditions. This variability can lead to inconsistent results and influence the diagnosis and treatment decision-making.

## 1. Introduction

Every year, shoulder dislocations occur on an average of 5–40 per 100,000 individuals in the general population [1,2,3,4,5]. Athletes and those that engage in sports activities are more prone to shoulder dislocation [6,7,8,9,10]. This is due to both the high repetitive loads involved in sports and traumatic events. A shoulder dislocation can damage the joint stabilizers and cause laxity. After a first dislocation, these individuals are more prone to redislocation events [4,11]. After repeated episodes of dislocation, these individuals can develop joint instability [4,12], which can result in persistent long-term pain and functional limitations. Therefore, an early diagnosis of shoulder instability is crucial in order to implement early treatment and secondary prevention strategies.

The diagnosis of shoulder instability is mostly based on clinical history and manual glenohumeral (GH) laxity tests [13] that are combined with imaging scans to assess the structural integrity of musculoskeletal structures [14]. Despite manual tests (apprehension and relocation tests) displaying high specificity, they only show suboptimal sensitivity, thus being non-optimal for identifying those with shoulder instability with high diagnostic accuracy [15,16]. Moreover, manual tests result in variable findings (e.g., inter-rater reliability) dependent on the experience, skill, and sensibility of the examiner. These tests can only subjectively evaluate the degree of shoulder instability and are unable to provide an accurate measurement of joint laxity.

To overcome the limitations of validity and replicability of manual laxity testing, a manifold of arthrometers have been developed to measure GH laxity. Joint arthrometers apply an external force to the joint with the aim of emulating the manual testing. Shoulder arthrometers are similar in concept to other arthrometers [16] for different joints that are already on the market (e.g., Telos, KT-1000/200). These devices can provide objective quantification and precise measurements of joint laxity leading to a more accurate diagnosis. Notwithstanding, the structure and use of these arthrometers can become heterogenous, i.e., inconsistent force application and patient positioning, and may lead to variable results and inconclusive findings. The use of arthrometers is important for clinical practice to provide a more precise estimate of GH joint laxity and to diagnose the presence and severity of shoulder instability. Clinical and arthrometric data on GH laxity would help clinicians to reach more accurate diagnosis and make an informed and adequate treatment planning (either conservative or surgical interventions). Scientific literature reporting shoulder arthrometry is still sparce and scattered, and there is no available source that systematizes these data, which can lead to inconsistent implementation of shoulder arthrometry and misinterpretation of their results. There is thus a clear need to systematize the scientific literature of the results of shoulder arthrometers for measuring GH laxity for consistent and reliable use of shoulder arthrometry. Our goal was to systematize the results of currently available shoulder arthrometers for measuring GH laxity in individuals with a healthy or injured shoulder. The purpose of this systematic review was thus to summarize state-of-the-art of shoulder laxity measurement using arthrometry and to compare the results across the different available devices and between injured and healthy shoulders. This summary of current evidence can guide researchers and clinicians on how to use arthrometers for measuring shoulder laxity and compare their results with available data according to the different characteristics of the shoulder condition(s), arthrometer used, and method of measurement.

## 2. Materials and Methods

This systematic review was conducted according to the Preferred Reporting Items for Systematic Reviews and Meta-Analyses (PRISMA) 2020 statement [17]. The protocol was a priori registered in the PROSPERO database under the number CRD42023404088.

### 2.1. Eligibility Criteria

The eligibility criteria are framed according to the Participants, Intervention/Exposure, Comparison, Outcome and Study design (PICOS) strategy.

As the eligible population, we included studies comprising both male and female individuals that were healthy (asymptomatic) and those presenting symptomatic shoulder conditions (shoulder pain and shoulder instability, among other conditions). Studies including individuals after shoulder stabilization surgery or rehabilitation were deemed eligible if they had the outcome of interest (GH laxity).

Under exposure and the outcome of interest, studies that made an arthrometer-assisted measurement of GH laxity were also considered. GH laxity was defined as the objective quantification of joint displacement (mm) or joint stiffness (N/mm). Arthrometers must include a system that measures joint laxity (either visually, as an external connected device or software, or with concomitant imaging methods). The external application of load had to be applied locally at the shoulder joint by a direct mechanical actuator (activated manually or by an electrical system). Studies where the load was applied via free weights with a pulley system were excluded. A comparator group of exposure was not compulsory, but studies comparing shoulder arthrometry with other methods were included. Studies involving different methods under the implementation of shoulder arthrometer evaluation (e.g., different populations, patient positioning, applied loads, and methods of measurement) were also considered.

We included all laboratory or clinical trials (from randomized controlled trials to case series) that allowed for the evaluation of the outcome of interest. Letters, editorials, conference abstracts, cadaveric and animal studies, case studies, commentaries, and reviews were excluded. Due to unavailable translation resources, only included studies written in English were analyzed.

### 2.2. Search Strategy

Computerized searches were conducted in the PubMed, EMBASE and Web of Science databases up to 3 May 2023. The full search strategy for each database is reported in Appendix A. The reference lists of the relevant reviews and of the included studies were screened for additional potentially eligible studies not identified via the database searches.

### 2.3. Study Selection

All records were exported to EndNote X7 (Thomson and Reuters, Philadelphia, PA, USA); duplicates were removed using the software’s “duplicates” tool and then confirmed manually to check for any missing duplicate records. Two authors (E.G. and R.A.) independently scanned all titles and abstracts, and then revised the full texts of all potentially eligible studies. Disagreements were resolved by a third author (C.V.).

### 2.4. Data Collection and Extraction

All data related to the study characteristics, arthrometers, and their outcomes were extracted in duplicate by two authors (E.G. and R.A.). Disagreements were resolved by consensus. We used an excel spreadsheet to record data, including: (i) Study characteristics (year and region); (ii) characteristics of the included population (number of individuals and shoulders, percentage of males/females, mean age, height, and body weight) and their clinical condition (asymptomatic or symptomatic); (iii) description of the arthrometer and testing conditions (name of the device, method of force application, amount of load, direction of force, patient positioning, shoulder fixation, laxity measurement system, and procedure) and their validity and reliability outcomes; (iv) outcome measures (laxity and stiffness).

### 2.5. Data Management

Studies that included overlapping populations but that presented different outcomes were merged into a cluster of studies. The population characteristics were collected for each study (as means and standard deviations), but then summarized using proportions and pooled means, as well as standard deviations weighted to the sample size.

When studies reported data for both shoulders or subgroups by sex, we collected and reported the outcomes of both shoulders/sexes separately (when available). When reported, the mean difference between shoulders was also collected. When studies presented data for the same outcome using different measuring methods (e.g., radiography and ultrasound) and varying loads or different patient positioning, the data from both methods in data synthesis were reported separately. However, when data were reported for the same population under the same testing conditions (e.g., for test–retest purposes) from the same study but from different trial reports, the data were combined using pooled means and standard deviations.

### 2.6. Risk of Bias

The risk of bias was judged using the Risk of Bias Assessment tool for Non-randomized Studies (RoBANS) [18]. The RoBANS is a validated tool to assess the risk of bias of non-randomized studies, comprising six domains of bias: (i) The selection of participants, (ii) confounding variables, (iii) measurement of exposure, (iv) blinding of outcome assessment, (v) incomplete outcome data, and (vi) selective outcome reporting (Appendix A). Each domain was judged as low risk of bias, high risk of bias, or unclear. Risk of bias was judged at the outcome level for laxity and stiffness. Two authors (E.G. and R.A.) made all judgements, and disagreements were resolved by a third author (C.V.).

### 2.7. Data Synthesis

Data were stratified according to population characteristics into three subgroups: (i) Healthy individuals with asymptomatic shoulders, (ii) athletes with asymptomatic shoulders, and (iii) individuals with injured shoulders.

Data pooling for joint laxity and stiffness was not attempted due to the wide heterogeneity across the studies’ populations, arthrometers, and testing methods. In addition to stratification based on population characteristics, we also stratified the data based on the device used and we organized data in regard to testing characteristics (e.g., measurement system, amount of load, shoulder positioning, and shoulder being assessed). Laxity data were then plotted into figures for visual display of the anterior (PA), posterior (AP), inferior, and global laxity. Stiffness was not reliable to plot into a figure due to many overlapping slopes, and it was thus reported for each individual study.

## 3. Results

The database and hand-searches yielded 2614 titles and abstracts. After removing the duplicates, the full texts of the 1435 relevant studies were analyzed according to the eligibility criteria. A total of 24 trial reports from 23 studies [19,20,21,22,23,24,25,26,27,28,29,30,31,32,33,34,35,36,37,38,39,40,41] met the eligibility criteria and were included in this systematic review (Figure 1).

### 3.1. Risk of Bias

Nearly one-fourth of the studies (k = 9; 23%) were judged as having a high risk of selection bias due to the selection of participants, including healthy individuals and athletes with asymptomatic shoulders but reporting shoulder pain or a previously diagnosed shoulder injury. More than half of the studies (k = 13; 57%) were judged as having a high risk of selection bias due to uncontrolled confounding variables, mostly due to unreported sex and unbalanced shoulder dominance. Three-quarters of the studies (k = 15; 75%) were judged as having a high risk of performance bias for their measurement of laxity, but none for measuring stiffness. Likewise, most of the studies were also judged as having a high risk of detection bias for their laxity (k = 13; 65%) and stiffness (k = 4; 80%) measurements. Incomplete data outcome was not a concern, with no study being judged as having a high risk of attrition bias. Only one study [32] was judged as having a high risk of selective reporting, but as no study registered an a priori protocol, this domain should be viewed with some concerns for all studies. The judgment of risk of bias for each included study and domain is provided in Appendix A.

### 3.2. Population Characteristics

A total of 1162 shoulders from 813 individuals with a mean age of 23.0 ± 3.7 years were included for analysis (Table 1). Among them, 401 were voluntary individuals with asymptomatic shoulders, 278 were athletes with asymptomatic shoulders, and 134 were individuals with injured shoulders. Appendix A details the population characteristics for each study.

### 3.3. Device Characteristics

Six different shoulder arthrometers were reported in the literature (Appendix A). The most reported arthrometers were the Telos + LigMaster™ (six studies) [27,28,30,34,35,36] and the customized instrumented shoulder arthrometer (six studies) [20,21,22,23,39,40], followed by the Telos (five studies) [24,25,26,29,37]. The remaining arthrometers included a shoulder adaptation of the KT-1000/2000 (three studies) [31,38,41], the Donjoy^®^ Laxity Tester (two studies) [32,33], and a custom-designed robotic device (one study) [19]. The validity and reliability data from these arthrometers are described in Appendix A.

### 3.4. Characteristics of Evaluation and Method of Measurement

Sensors were the most commonly used method for measuring GH laxity and stiffness (15 studies). The sensors measured the force-induced position changes at the glenohumeral joint, which were used calculate the joint laxity and/or stiffness (12 studies) [20,21,22,23,27,28,30,34,35,36,39,40], while the other devices visually displayed the amount of laxity on a screen (three studies) [31,38,41]. Apart from sensors, other studies used stress imaging either with radiography or US devices (five studies) [24,25,26,29,37], a visual scale (two studies) [32,33], or a digital motion controller (one study) [19].

Force was applied with a controlled manual instrumented-assisted mechanism, with only one study using an electromechanical system [19]. The direction of force was usually in the sagittal plane (AP or PA), with only three studies applying an inferior-directed force [19,22,23]. The amount of load applied during the testing procedures was heterogenous across the studies. The load applied ranged from 10 to 150 N. Three studies used a progressive application of load (0–100, 0–134, or 10–80 N) and two studies applied force until the capsular endpoint [22,23].

The shoulder was usually positioned at 90° of abduction in the scapular plane (15 studies) [19,24,25,26,27,28,29,30,31,34,35,36,37,38,41], in either external rotation (12 studies) or neutral rotation (four studies). Other studies positioned the shoulder at 20° of abduction with neutral rotation (nine studies) [20,21,22,23,32,33,39,40,41]. The individuals were either lying in a supine position (modified KT-1000/2000 and modified custom-designed robotic device) or seated in a chair (customized instrumented shoulder arthrometer, Telos GA-II/E, Telos, and Donjoy^®^). Methods of fixation varied considerably across the studies and devices, and are detailed in Appendix A.

The measurement methods for GH laxity showed heterogeneity across the included studies (Appendix A). Imaging methods measured the bone displacements (distance between the center or posterior humeral head to the glenoid) to calculate the joint laxity. When using sensors, laxity was calculated as the distance between two sensors placed at the humeral head and acromion (sagittal displacement) or between the humeral head and lateral epicondyle of the distal humerus (inferior displacement). While in most studies it was measured to total displacement, other studies restricted the displacement to the distance between the inflection point until the data point at the highest load. Stiffness was always calculated as the slope of the linear portion of the force–displacement curve. The KT1000/2000 estimated the humeral head displacement with a single sensor placed at this anatomical point, but without any other reference point. The Donjoy^®^ arthrometer used a visual-instrumented scale on the spring balance to measure the sagittal displacement of the humeral head.

### 3.5. Laxity and Stiffness Values

The GH laxity was analyzed across asymptomatic healthy individuals, asymptomatic athletes, and individuals with injured shoulders. Overall, there was a wide heterogeneity in laxity values across and within subgroups of individuals with asymptomatic shoulders, especially when compared between devices. The laxity and stiffness data for each study and testing condition are detailed in Appendix A.

Asymptomatic healthy individuals showed varying laxity values, ranging from 0.7 to 27.72 mm for PA, 1 to 21.75 mm for AP (Figure 2a), and 0.6 and 2.1 mm for global translation (Figure 2b). Stiffness ranged from 16.3 to 16.7 N/mm for PA and 1.51 to 15.7 N/mm for inferior (Table 2). Only one study [22] assessed inferior laxity with a 13.9 mm displacement, while another study assessed AP stiffness with a similar slope of 15.4 N/mm [23].

Asymptomatic athletes also showed varying laxity values, ranging from 1.4 to 12.57 mm for PA, 4.82 to 12.71 mm for AP (Figure 3a), and 7.81 to 24.92 mm for global translation (Figure 3b). When comparing the dominant/throwing arm against the contralateral, there was no significant difference in the mean laxity values. The mean values of stiffness also showed a large range from 7.77 to 16.6 N/mm for PA and 8 to 15.3 N/mm for AP (Table 2).

Injured shoulders displayed a more uniform pattern of laxity, with mean values ranging from 2 to 3.4 mm for PA, 3.0 to 5.42 mm for AP (Figure 4a), and 2.8 to 11.9 mm for global translation (Figure 4b). None of the included studies assessed the stiffness of individuals with injured shoulders.

## 4. Discussion

The most important finding of this systematic review is the high diversity of GH laxity outcomes. This considerable variation might have resulted from the wide variety of arthrometers being used and their different testing conditions, such as the amount of load, shoulder positioning, and methods of measurement.

### 4.1. Why Are the Laxity Outcomes So Variable across Studies?

The Telos device, which comes in different device models and adaptations, is the most widely used arthrometer and is always used in conjunction with either imaging methods (radiography or ultrasound) or the Ligmaster™ for measuring GH laxity or stiffness. The Ligmaster™ is a software incorporated into the Telos device to allow the estimation of joint laxity without the need of imaging control. This software accounts for soft tissue compression by calculating an inflection point along the force–response curve, providing an approximation of GH displacement. In contrast, the Telos device, when used concomitantly with imaging control, can provide an adequate measurement of GH laxity by means of measuring the true bone displacements before and after stress. Other available arthrometers (custom-designed robotic device, customized instrumented shoulder arthrometer, modified KT-1000/2000, and Donjoy^®^) can measure the GH laxity without concomitant imaging control, and thus do not provide accurate laxity measurements. More specifically, the modified KT-1000 and KT-2000 usually result in considerably higher GH laxity values when compared to the other arthrometers. The rationale behind this may be that this device was designed with sensors that are not sensitive to the displacement of the humeral head.

Varying laxity outcomes can also arise from heterogeneity in the testing conditions. Most arthrometers apply an external force to the humeral head or superior humerus by a manually assisted mechanism. Although the arthrometer may provide a system to control the manually assisted forces applied (usually a dynamometer), there is nevertheless some risks for poor reproducibility in repeated measures. Laxity was commonly evaluated in the sagittal plane, but with varying loads, usually ranging from 67 to 150 N. Only one study measured and reported the inferior GH laxity (axial plane) [22]. The reduced number of studies evaluating the inferior shoulder translations may be due the fact that inferior shoulder dislocations are uncommon [42]. The shoulder positioning was less heterogenous across studies, with the shoulder usually being positioned at 90° of abduction in the scapular plane, but perhaps varying in the positioning of rotation (neutral or external rotation).

### 4.2. Is GH Laxity Symmetric and What Are the Differences between Symptomatic and Asymptomatic Shoulders?

Most shoulder dislocations occur anteriorly (85–95%) [43,44] and thus higher mean values of anterior GH laxity were expected, but the studies did not show this tendency. The mean values for anterior laxity (as evaluated by Telos and Telos + Ligmaster™) were not relevantly different between athletes and non-athletes with asymptomatic shoulders. More relevant differences were only seen for posterior laxity, with athletes showing higher laxity mean values. Several sports require athletes to repeatedly perform overhead movements, placing large intra-articular force and stress on their shoulders, which can result in GH laxity and instability [45,46,47]. This mechanism results in some asymptomatic athletes displaying asymmetrical shoulder laxity that may eventually develop instability-related symptoms (e.g., pain) [48,49,50]. Excessive laxity, mostly due to repetitive microtrauma or functional deterioration, can cause functional limitations and force many of these athletes to sit out of games [51]. Notwithstanding, the included studies reported no statistically significant differences between throwing and non-throwing arms and differences were small and probably clinically irrelevant [24,28,29,30]. There was a wide variation in stiffness values among the different studies, without clinically relevant differences between throwing and non-throwing arms or between anterior and posterior stiffness. These findings do not support the evidence that the throwing arms of athletes commonly develop a thickened and fibrotic posterior capsule [52,53,54,55]. However, some caution is required when interpreting the stiffness outcomes (especially if comparing them to cadaveric studies) because all of the included studies calculated stiffness by means of determining the slope of the linear section of the force–displacement curve within the elastic region, which could potentially lead to an overestimation of these values.

The anterior and posterior mean GH laxity values were not different between injured and non-injured shoulders when measured solely by the Telos device. This finding is interesting because it was expected that injured shoulders would present higher sagittal GH laxity due to the nature of the injury (anterior dislocation or multidirectional instability), which is associated with higher grades of GH laxity [56]. However, when the GH laxity was compared in the same study using exactly the same testing conditions, the anterior laxity (and not posterior) was significantly superior in shoulders with instability [37]. The same findings were not seen when comparing shoulders with a history of shoulder pain (due to tendinitis, thoracic outlet syndrome, or labral tear) to healthy asymptomatic shoulders, but these injury conditions are not prone to cause either shoulder instability or GH laxity [26].

### 4.3. Recommendations for Clinical Practice and Future Directions

Assessing GH laxity with manual testing is simple, accessible, and practical. However, the clinician should be aware of the poor reproducibility, insufficient reliability, and limited precision of these manual tests, which do not allow to accurately measure laxity, but solely provide a subjective assessment of joint laxity. The use of arthrometers to objectively quantify GH laxity is important in order to overcome the abovementioned limitations of manual laxity tests [57,58].

Although arthrometers are widely used to evaluate the laxity of knee and ankle joints [59,60,61], there are still a limited number of studies and available arthrometers to measure GH laxity. The process of developing a shoulder arthrometer device is challenging due to the complex interactions between the GH and other shoulder joints, the difficulty in measuring changes in translation due to soft tissue compliance, and the individual’s inability to relax [16]. Most arthrometers rely on sensors to estimate the GH laxity, but this method is less precise because it is not able to measure true intra-articular bone displacement. When available, clinicians should prioritize those arthrometers that allow to confirm bone displacement with imaging control.

Although heterogeneity was found in the testing characteristics, there are still clinically relevant findings that can be summarized. When comparing GH laxity values to those reported in the literature, researchers and clinicians must be aware that GH laxity varies considerably across arthrometers and should thus compare their results to those that are specific to the arthrometer being used. Shoulder position (especially related to rotation) can affect the GH laxity values and clinicians should be aware of the potential differencing factor. The shoulder may be evaluated in both positions (neutral and external rotation) or the position that best applies to the population being assessed (for example, in external rotation for overhead athletes). The devices also offer an array of different possible loads, but it appears that loads of around 100–150 N are the most common for sagittal translation. For stiffness measurements, a progressive and controlled increment of loading is required to compute the force–displacement curves.

Future research should aim to standardize arthrometer devices and the methods for measuring GH laxity. Until then, there is no reliable comparison of GH laxity values across studies, and thus no definitive conclusions can be made in regard to normal physiological laxity and pathological laxity. A standardized device and methods of measurement are in great need. These would allow to calculate cut-off values to assist the medical community in deciding on the most effective treatment for each condition. Devices that allow concomitant imaging control should also be prioritized in order to allow bone displacement measurements. Arthrometers that can be used concomitantly with magnetic resonance imaging are already being used for other joints [62,63,64,65,66,67,68] and they allow to correlate the structural integrity of anatomical structures (most notably the ligaments) with the joint’s functional competence (laxity). Future directions may focus on the development of a similar device that can be applied to the shoulder joint in a similar fashion.

### 4.4. Limitations

This systematic review has several limitations that should be considered when interpreting the results. The studies included in this systematic review exhibited variability in the devices and methods used to objectively measure GH laxity, and only one study [19] used an electromechanical device to apply the force during the test. Other devices applied the load with a hand-assisted mechanism, which may have resulted in less reproducible loads being applied to the joint. The differences between the various devices resulted in inconsistent GH laxity outcomes when comparing healthy individuals or athletes with asymptomatic shoulders to individuals with injured shoulders. The variability found in laxity and stiffness values did not allow to estimate reliable cut-offs to differentiate physiological and pathologic GH laxity. Moreover, the absence of validity and reliability data for arthrometers in some studies [21,24,26,30,33,34,35,36] makes it uncertain if these devices provide consistent and accurate measurements.

## 5. Conclusions

There is a wide heterogeneity of GH laxity outcomes across the literature, which may be explained by the different arthrometer devices used. This limitation hampers the generalizability of our findings and a reliable estimation of physiologic and pathologic GH laxity. Most arthrometers use sensors to measure joint displacement, but these are prone to less accurate and imprecise laxity measurements. The use of concomitant imaging control should be considered the “gold standard” method to calculate bone displacement and measure the GH laxity. Future research should focus on standardization of the use of shoulder arthrometers and their methods to strive for a valid comparison of results across the literature and to allow to determine reliable cut-offs that can be used to assist clinical decisions.

## Figures and Tables

**Figure 1 bioengineering-10-00799-f001:**
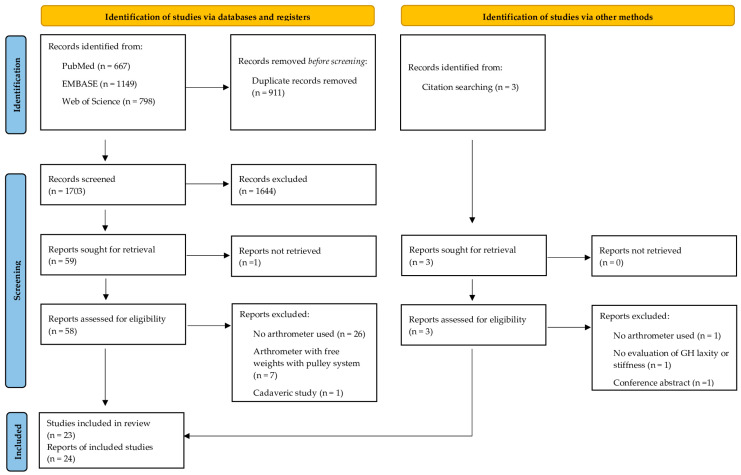
PRISMA 2020 flow diagram for new systematic reviews that included searches of databases, registers, and other sources.

**Figure 2 bioengineering-10-00799-f002:**
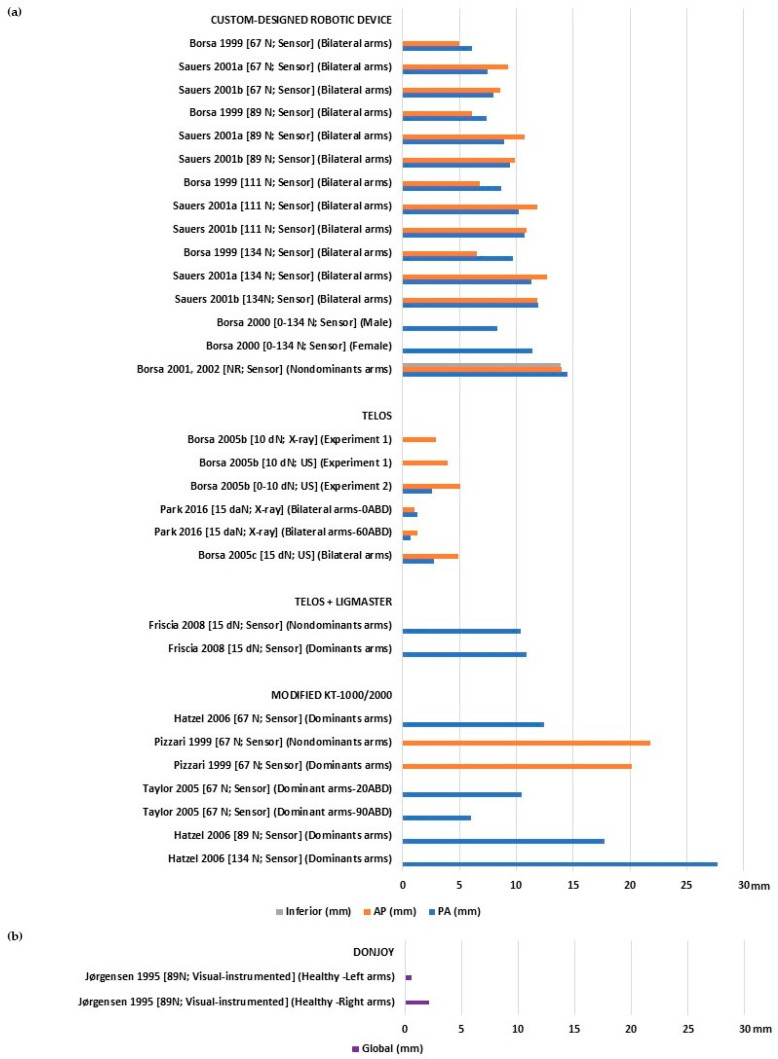
Glenohumeral laxity measurement outcomes of healthy individuals with asymptomatic shoulders: (**a**) Anterior, posterior, and inferior translation [20,21,22,23,25,26,30,31,37,38,39,40,41]; (**b**) global sagittal translation [32].

**Figure 3 bioengineering-10-00799-f003:**
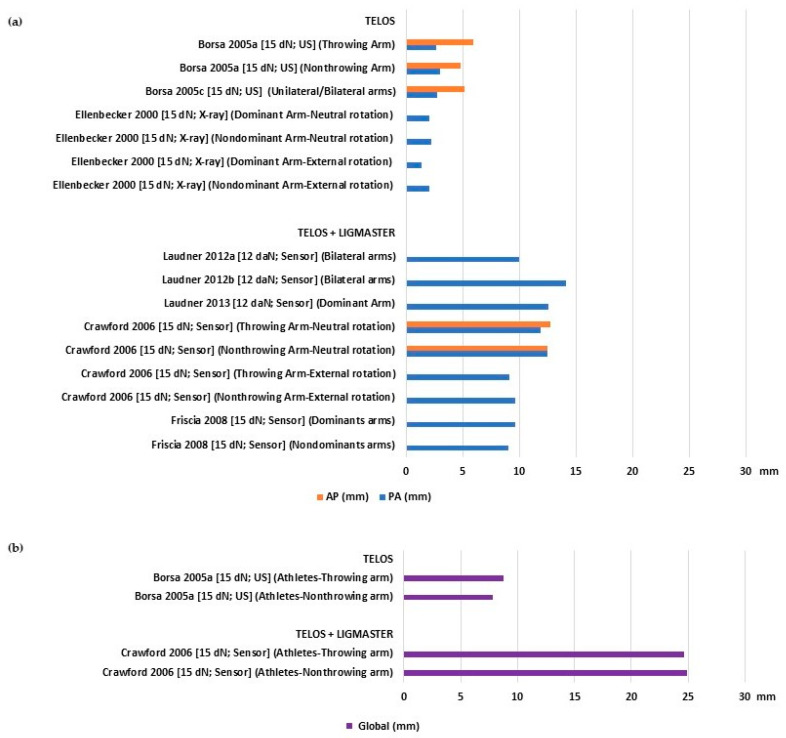
Glenohumeral laxity outcomes of athletes with asymptomatic shoulders: (**a**) Anterior and posterior translation [24,26,28,29,30,34,35,36]; (**b**) global sagittal translation [24,28].

**Figure 4 bioengineering-10-00799-f004:**
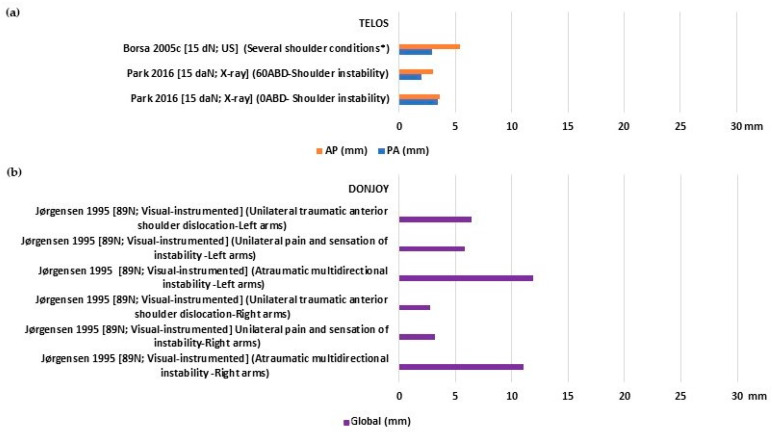
Glenohumeral laxity outcomes of individuals with injured shoulders: (**a**) Anterior and posterior translation [26,37]; (**b**) global sagittal translation [32]. * Tendinitis/Thoracic outlet syndrome/Labral tear.

**Table 1 bioengineering-10-00799-t001:** Summary of the population characteristics of the included studies.

Variable	Asymptomatic Shoulders		Injured Shoulders	Total Sample
	Healthy Individuals	Athletes		
K (population)	401	278	134	813
N (shoulders)	570	455	137	1162
Sex (M/F)	196/161	103/41	71/36	370/238
Age (years)	21.2 ± 7.2	21.5 ± 2.9	24.4 ± 7.7	23.0 ± 3.7
Weight (kg)	72.9 ± 6.1	83.6 ± 13.2	NR	79.3 ± 12.1
Height (cm)	171.7 ± 4.3	182.0 ± 7.7	NR	177.8 ± 8.5

NR: No reported.

**Table 2 bioengineering-10-00799-t002:** Stiffness of the included studies.

Population	Arthrometer	Studies	Amount of Load	Device	Evaluated Arms	PA	AP	Inferior
**Healthy Individuals**	Custom-designed robotic device	Azarsa et al. (2021) [19]	10–80 N	Digital motion controller + software	Right arm	NR	NR	1.51
	Customized instrumented shoulder arthrometer	Borsa et al. (2001,2002) [22,23]	NR	Sensor	Nondominant arms	16.7	15.4	15.7
		Borsa et al. (2000) [21]	0–134 N	Sensor	Bilateral arms—male	20.5	NR	NR
		Borsa et al. (2000) [21]	0–134 N	Sensor	Bilateral arms—female	16.3	NR	NR
**Healthy Athletes**	Telos + Ligmaster	Crawford & Sauers (2006) [28]	15 dN	Sensor	Throwing arm—neutral rotation	8.05	8.00	NR
		Crawford & Sauers (2006) [28]	15 dN	Sensor	Nonthrowing arm—neutral rotation	7.77	8.05	NR
		Crawford & Sauers (2006) [28]	15 dN	Sensor	Throwing arm—external rotation	10.87	NR	NR
		Crawford & Sauers (2006) [28]	15 dN	Sensor	Nonthrowing arm—external rotation	10.24	NR	NR
		Borsa et al. (2006) [27]	15 dN	Sensor	Throwing arm	16.6	15.1	NR
		Borsa et al. (2006) [27]	15 dN	Sensor	Nonthrowing arm	16.2	15.3	NR

## Data Availability

Publicly available datasets were analyzed in this study. These data can be found in the References section.

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
