# Peer review of "Inconsistency in Shoulder Arthrometers for Measuring Glenohumeral Joint Laxity: A Systematic Review"

_bioengineering, 2023, doi:10.3390/bioengineering10070799_

Round 1

Reviewer 1 Report

The authors describe a review on the glenohumeral joint laxity that leads to shoulder inconsistency. They based their study on clinical data reported in 23 articles present in different databases. More than 1000 shoulder, comprising symptomatic and asymptomatic individulas both athletes and non athletes, were examined comparing a possible correlation between the arthrometers used and the glenohumeral laxity measurements.

Although it is not an experimental work, the manuscritpt reports a detailed overview of this pathological condition. The authors conclude that there is a wide heterogeneity in the data on shouder laxity reported in the leterature depending on the arthrometers used. They propose some clinical reccomendations for future researches. 

The manuscript appears well written, discussed and refered. 

The only minor point that I note is the number of shoulders considered. As reported in the abstract and in the results (3.2 population characteristics) and in table 1, the total  is often different from the sum. For example, in the abstract there are 1162 shoulders that does not correspond to the sum of 401 + 278 + 137. Similarly in the other sections. The authors should be more precise in the description by well distinguishing the categories.

Author Response

Dear reviewer,

We would like to express our gratitude for your time in reviewing our manuscript and providing helpful commentary. Please see below our replies to your comments.

REVIEWER COMENT: The authors describe a review on the glenohumeral joint laxity that leads to shoulder inconsistency. They based their study on clinical data reported in 23 articles present in different databases. More than 1000 shoulder, comprising symptomatic and asymptomatic individulas both athletes and non athletes, were examined comparing a possible correlation between the arthrometers used and the glenohumeral laxity measurements.

Although it is not an experimental work, the manuscritpt reports a detailed overview of this pathological condition. The authors conclude that there is a wide heterogeneity in the data on shouder laxity reported in the leterature depending on the arthrometers used. They propose some clinical reccomendations for future researches. 

The manuscript appears well written, discussed and refered.

The only minor point that I note is the number of shoulders considered. As reported in the abstract and in the results (3.2 population characteristics) and in table 1, the total is often different from the sum. For example, in the abstract there are 1162 shoulders that does not correspond to the sum of 401 + 278 + 137. Similarly in the other sections. The authors should be more precise in the description by well distinguishing the categories.

AUTHORS’ REPLY: Thank you again for your time and for catching our typo. We have revised this at the abstract and in results text, from 137 to 134 individuals with injured shoulders.

Reviewer 2 Report

The paper was well written. The study was well constructed and the results interpreted correctly.

Author Response

Dear reviewer

Thank you very for your time reviewing our submission and for your thoughtful comments.

Reviewer 3 Report

1.      The introduction is very well grounded, but please emphasise the elements of originality/ novelty that this study brings to the specialised literature. I understand that Our goal is to systematise and summarise the results of currently available shoulder arthrometers for measuring GH laxity in healthy and injured individuals., so please:

·        Please rephrase and better highlight the purpose of this study;

·        Please specify exactly what is new in your study or where it stands in the scientific literature.

2.      Attached is the similarity coefficient ratio, which is within normal parameters.

3.      Congratulations to the authors for their hard work in developing this systematic review.

Minor editing of the English language is required.

Author Response

Dear reviewer,

Thank you for your time in reviewing our submission and for the helpful comments that enhanced the quality of our paper. Please see below the point-by-point replies to your comments.

REVIEWER COMMENT: 1. The introduction is very well grounded, but please emphasise the elements of originality/ novelty that this study brings to the specialised literature. I understand that Our goal is to systematise and summarise the results of currently available shoulder arthrometers for measuring GH laxity in healthy and injured individuals., so please:

  • Please rephrase and better highlight the purpose of this study;
  • Please specify exactly what is new in your study or where it stands in the scientific literature.

AUTHORS’ REPLY: Thank you for your comment and suggestions. We understand that our purpose and rationale for the study may not be so clear; therefore, we have revised both these features at Introduction. We added a few lines at the Introduction (before the objective) to provide readers on the rationale for our systematic review: why we made it, its novelty (there is no other source that systematizes this data) and why our work is important. Regarding our purpose (goal), we have not edited much because, as per PRISMA 2020 guidelines, the purpose should be framed according to the PICO strategy. However, we added a few additional sentences to highlight and expand on the purpose of our systematic review and how our results can be useful to guide clinicians that want to implement shoulder arthrometers in the clinical practice.

REVIEWER COMMENT: 2.  Attached is the similarity coefficient ratio, which is within normal parameters.

AUTHORS’ REPLY: Thank you for checking our submission for risk of plagiarism. We checked the results of the report and we saw that most of similarities are regarding to isolated words/expressions and non-relevant data (references or affiliations). Therefore, we feel that there is no need for further changes for these purposes, and assumed from your comment that none are requested.

REVIEWER COMMENT: 3. Congratulations to the authors for their hard work in developing this systematic review.

AUTHORS’ REPLY: We are very thankful for your thoughtful comments. 

REVIEWER COMMENT: Minor editing of the English language is required.

AUTHORS’ REPLY: We thank you for the time you invested to provide feedback on our manuscript. We have re-read our manuscript, which was proof-read by an English native speaker.